# Evaluation of the Antihypertensive Activity of Eggplant Acetylcholine and γ-Aminobutyric Acid in Spontaneously Hypertensive Rats

**DOI:** 10.3390/molecules28062835

**Published:** 2023-03-21

**Authors:** Wenhao Wang, Shohei Yamaguchi, Masahiro Koyama, Kozo Nakamura

**Affiliations:** 1Department of Science and Technology, Graduate School of Medicine, Science and Technology, Shinshu University, 8304 Minamiminowa, Nagano 399-4598, Japan; 2Wellnas Co., Ltd., 1-28-5 Koenjiminami, Suginami-Ku, Tokyo 166-0003, Japan; 3Department of Bioscience and Biotechnology, Faculty of Agriculture, Shinshu University, 8304 Minamiminowa, Nagano 399-4598, Japan; 4Institute of Agriculture, Academic Assembly, Shinshu University, 8304 Minamiminowa, Nagano 399-4598, Japan

**Keywords:** eggplant, acetylcholine, γ-aminobutyric acid, spontaneously hypertensive rat, hypertension, functional food

## Abstract

Daily consumption of eggplant powder containing 2.3 mg acetylcholine (ACh) is known to alleviate hypertension and improve mental status. However, eggplant powder used in clinical trials also contains the antihypertensive compound γ-aminobutyric acid (GABA). Although our previous study indicated that the main antihypertensive compound in eggplant is ACh, given that GABA amounts in eggplant do not reach the effective dosage, the effects of GABA on the antihypertensive effect of eggplant remain unclear. It is necessary to establish whether there is a synergistic effect between GABA and ACh and whether GABA in eggplant exerts antihypertensive effects. Consequently, here we sought to evaluate the effects of GABA on the antihypertensive effects of eggplant. We used a probability sum (*q*) test to investigate the combined effects of ACh and GABA and prepared eggplant powder with very low ACh content for oral administration in animals. ACh and GABA exhibited additive effects but the GABA content in eggplants was not sufficient to promote a hypotensive effect. In conclusion, ACh is the main component associated with the antihypertensive effects of eggplant but GABA within eggplants has a minimal effect in this regard. Thus, compared with GABA, ACh could be a more effective functional food constituent for lowering blood pressure.

## 1. Introduction

Hypertension is a prominent risk factor that increases the likelihood of developing brain, heart, kidney, and other diseases [1]. Worldwide, an estimated 1.13 billion individuals are living with hypertension [2]. The main causes of hypertension are a combination of genetic and environmental factors, among the latter of which are diet, lack of exercise, smoking, and alcohol consumption. Research on improving blood pressure via dietary modification has received increasing attention in recent years, particularly in the context of antihypertensive nutraceuticals and functional foods [3]. Consequently, the discovery of a broader range of functional foods with antihypertensive properties could contribute to diversifying the dietary options and thus contribute to enhancing the health status of hypertensive individuals.

Eggplant (*Solanum melongena* L.), a common vegetable consumed daily worldwide, is an abundant source of minerals and dietary fiber; contains vitamins B_6_, C, and B_9_; and is low in protein and calories [4]. Furthermore, eggplants contain functional compounds such as chlorogenic acid, which has anti-inflammatory and antioxidant properties [5,6]; γ-aminobutyric acid (GABA), with antihypertensive effects [7,8]; and nasunin with anti-angiogenic activity [9,10].

Acetylcholine (ACh) is an ester compound comprising quaternary ammonium cation choline and acetic acid that functions as a neurotransmitter, not only in the nervous system of mammals but also in that of many other organisms [11,12]. However, given that orally administered ACh is rapidly broken down by cholinesterase in the body and thus not readily absorbed, it has long been used as an intravenous drug. In this regard, we have previously conducted single and repeat oral studies of dried eggplant powder using spontaneously hypertensive rats (SHRs) and confirmed a significant antihypertensive effect. We speculate that the substance promoting the hypotensive effect is ACh [13], the content of which in eggplant was found to 2900-fold higher than that in other cultivated crops [14]. This finding confirmed, for the first time, that orally administered ACh has potential utility as a novel functional food compound. On the basis of the results of animal studies, we conducted a randomized, placebo-controlled trial on hypertensive patients and accordingly found that the daily intake of ACh-rich eggplant powder (ACh content: 2.3 mg) contributed to a significant reduction in blood pressure and enhanced mental status (stress and mood) [15]. The eggplant powder used in the clinical trial contained GABA in addition to ACh. The daily GABA intake in this clinical trial was 7.65 mg [15], which is lower than the previously reported effective dose of 10 mg [7], although GABA may have been regarded to be a common antihypertensive compound in lower effective dosages [7,16,17]. On the basis of these findings, we assume ACh to be the major antihypertensive factor in eggplant powder. However, we have yet to sufficiently establish the role of GABA in the antihypertensive effect of eggplant consumption, i.e., whether GABA acts synergistically with ACh or whether the GABA content in eggplant is unassociated with the hypotensive effect. Accordingly, establishing the contribution of GABA in this regard is important with respect to determining whether the minimum effective intake of eggplant should take only the ACh content into account, which would reduce the cost of testing the contents of different functional compounds in products during the commercialization of eggplant products.

Consequently, in this study we sought to evaluate the effects of eggplant GABA on the antihypertensive effect of eggplants. We prepared an eggplant powder with low ACh content and the same level of GABA as in the eggplant powder used in the previous clinical trial and investigated the effects of oral administration to evaluate the blood pressure-lowering effects. In addition, we evaluated the interaction between ACh and GABA on the blood pressure-lowering effect in SHRs using the probability sum (*q*) test, a common analytical approach used to determine the combined effect of two medications. On the basis of the findings of this study, we identified the main factor in eggplants that exerts hypotensive effects and established the association between the GABA content of eggplants and it antihypertensive effects. Moreover, we highlight the merits of ACh compared with GABA as a hypotensive constituent in eggplants.

## 2. Results

### 2.1. Combined Effects of ACh and GABA

To determine whether GABA influences the hypotensive effect of ACh, we assessed the combined effects of ACh and GABA using the probability sum (*q*) test, the results of which obtained for rats treated with ACh, GABA, or a combination of ACh and GABA are shown in Table 1. At 6 h post-administration, rats characterized by a reduction in systolic blood pressure (SBP) of more than 20 mmHg compared with that prior to administration (0 h) were defined as responders. Of the six rats in the ACh group, two were found to have reductions in SBP of more than 20 mmHg, (i.e., a probability of *P_ACh_* = 2/6 = 1/3). Similar responses were recorded among the GABA group rats (*P_GABA_* = 2/6 = 1/3), whereas three of the six rats in the ACh + GABA group showed a reduction in SBP greater than 20 mmHg (*P_ACh_*
_+ *GABA*_ = 3/6 = 1/2). Substituting the above probabilities into the *q* test equation [*q* = *P_ACh_*
_+ *GABA*_/(*P_ACh_* + *P_GABA_* − *P_ACh_* × *P_GABA_*)] yielded a *q*-value of 0.90. We accordingly established that the combination of ACh and GABA had an additive effect on the hypotensive effect in experimental rats, which tends to indicate an absence of any synergistic effect between GABA and ACh with respect to lowering blood pressure.

### 2.2. Quantification of the ACh and GABA Contents in Samples

To determine whether the GABA contained within eggplants has a hypotensive effect, we prepared eggplant powder with a very low ACh content (whereas the GABA concentration remained unmodified). ACh and GABA concentrations in the ACh-reduced eggplant powder (ACh-reduced eggplant sample) and the normally prepared eggplant powder (eggplant control sample) are shown in Table 2. The GABA content of the ACh-reduced eggplant sample was comparable to that of the control sample, whereas the ACh content was only 0.22% of that of the control sample. The doses of the ACh-reduced eggplant samples used in the oral administration studies could, therefore, be adjusted to give the same GABA dosage in both samples, whereas the ACh dosage delivered in the ACh-reduced eggplant sample was lower than in the eggplant control sample. Furthermore, to confirm the antihypertensive effects of GABA in eggplants, we subjected rats to three treatments, namely, the ACh-reduced eggplant preparation, positive control (eggplant control sample), and negative control (pure water).

### 2.3. Acute Antihypertensive Effect

The changes in the SBP of rats before and after the oral administration of eggplant powder suspensions are shown in Figure 1. Compared with rats in the negative control group (pure water), those in the positive control group (eggplant control sample) showed a declining trend of SBP at 3 and 6 h post-administration and significantly lower values at 9 h (*p* < 0.05), thereby confirming the short-term hypotensive effect of the eggplant control sample. Comparatively, compared with rats in the negative control group, those in the subject group (ACh-reduced eggplant sample) showed no significant difference following administration the eggplant preparation. Furthermore, fluctuations in the SBP of subject group rats were similar to those observed in the negative control group rats and no hypotensive effect was detected.

### 2.4. Chronic Antihypertensive Effect

Among rats in the positive control and subject groups, we detected no significant differences with respect to food intake, water intake, or body weight compared with those in the negative control group (Figure 2a–c). Figure 2d depicts the sequential changes in blood pressure, which indicates that the blood pressure of subject group rats (ACh-reduced eggplant sample) did not differ significantly from that of rats in the negative control group (pure water). Contrastingly, compared with the negative control groups rats, those in the positive control group (eggplant control sample) showed significantly lower values on days 14 (*p* < 0.05) and 21 (*p* < 0.01) following the commencement of administration, thereby confirming the inhibitory effect of eggplant on blood pressure elevation. These results thus provide evidence to indicate that the GABA contained in eggplant samples does not have any pronounced antihypertensive effect. Overall, the condition of rats in the treatment was good during the repeated oral administration study.

## 3. Discussion

In this study, we investigated the combined effects of ACh and GABA to determine the effect of GABA on the antihypertensive properties of eggplant. On the basis of our findings in a previous study, we had speculated that ACh is probably the main antihypertensive constituent in eggplants and that the content of GABA within eggplant is in all likelihood insufficiently high to have any appreciable effects in this regard [13]. To confirm this inference, we prepared ACh-reduced eggplant powder and indeed verified that the main antihypertensive compound in eggplant is ACh.

The results of our combined effect study revealed that the combination of ACh and GABA had an additive effect on the reduction in blood pressure in SHRs. The mechanisms underlying the observed reductions in blood pressure in response to the oral administration of ACh and GABA are assumed to be as follows. Following administration, ACh acts on the M_3_ muscarinic ACh receptor in the gastrointestinal tract without being absorbed, thereby promoting afferent vagal (parasympathetic) nerve activity, resulting in a reduction in sympathetic nervous activity associated with an autonomic reflex, suppression of noradrenaline (NAD) release, and lowering of acute and chronic blood pressure [18]. Autonomic reflexes are believed to involve afferent vagal (parasympathetic) stimulation transmitted to the rostral lateral medulla (RVLM), thereby inhibiting RVLM activity and thus a reduction in RVLM stimulation of sympathetic nerve activity [19,20]. In contrast, orally administered GABA is absorbed by the body [21], wherein it binds to GABA_b_ receptors in ganglionic or presynaptic (peripheral nerves) sites. In response to the activation of GABA_b_, K^+^ channels open, thereby bringing neurons closer to a K^+^ equilibrium potential, and as a consequence, there are reductions in both action potential frequency and neurotransmitter release [22]. This accordingly has the effect of attenuating sympathetic conduction, inhibiting NAD secretion, and reducing blood pressure [23,24]. On the basis of these putative blood pressure-lowering mechanisms of orally administered ACh and GABA, it is assumed that ACh does not influence the binding of GABA to the GABA_b_ receptor at ganglionic sites or presynaptic (peripheral nerves), and thus the oral administration of ACh is considered unlikely to enhance the sympathoinhibitory effects of GABA. As the blood-brain barrier is impermeable to GABA [25], orally delivered GABA is unable to enter the central nervous system and has no appreciable effect on the hypotensive effect of oral ACh. In summary, on the basis of the established mechanistic activities, it is assumed that orally administered ACh and GABA do not mutually enhance their respective sympathetic depressant effects, which explains our findings that the combined effects of these two compounds are additive. In this regard, it has previously been found that the effects of phenobarbital and spironolactone combination therapy on UDP-glucuronosyltransferase are additive, as these two compounds act via independent mechanisms [26]. Similarly, combination therapy using pravastatin and olmesartan has been shown to have additive effects in reducing myocardial infarction via different mechanisms [27]. These examples thus indicate that the additive effects of combined medications are probably attributable to the independent mechanisms of the individual preparations. Accordingly, we reason that our findings revealing that the combined effects of ACh and GABA are additive rather than synergistic provide evidence to indicate that the GABA contained within eggplants does not influence the antihypertensive effect of ACh.

In this study, we prepared a lyophilized powder of ACh-reduced eggplant to determine the antihypertensive effect of GABA in eggplant. In the preparation of the ACh-reduced eggplant samples, in which eggplant samples were stored frozen at −20 °C for 15 months after harvest, ACh was reduced to a level that is 0.22% of that in a corresponding eggplant control sample, which was prepared by freeze-drying immediately after harvest. The reduction in the ACh content of eggplant during long-term frozen storage at −20 °C is believed to be associated with the gradual degradation of ACh by cholinesterase, which is naturally present in eggplants [28].

For the single oral dose study, we established three experimental groups, namely, a subject group (the ACh-reduced eggplant sample), positive control group (the eggplant control sample), and negative control group (pure water), and the hypotensive effects of ACh-reduced eggplant or eggplant control sample were evaluated based on a comparison with the negative control group (pure water). The subject group showed values similar to those of the negative control group, whereas the positive control group confirmed the short-term hypotensive effect of eggplant, as demonstrated in a previous study [13].

In the repeated oral administration study, rats were administered either ACh-reduced or unmodified eggplant preparations over the mid- to long-terms and evaluated in vivo by comparing the changes in general conditions and blood pressure with those of rats in the negative control group. Rats in the subject group were found to show blood pressure changes comparable to those recorded in the negative control group rats, whereas measurements obtained for the positive control group rats confirmed the mid- to long-term inhibition of blood pressure increases that occur during aging, as previously demonstrated [13].

In contrast to the significantly lower levels of ACh in ACh-reduced eggplant samples, the GABA content in these samples was similar to that in eggplant control samples. Our studies on SHRs orally administered the ACh-reduced eggplant and eggplant control samples revealed that blood pressure was not lowered in the subject group rats and only those in the positive control group showed a hypotensive effect, thereby providing evidence that ACh is the main hypotensive substance in eggplants and that GABA in eggplant has no apparent hypotensive effect at the assessed dose. The blood pressure-lowering low dose of GABA for SHRs is reported to be 19 μmol/kg body weight (BW) [29], which would explain our findings indicating that the ACh-reduced eggplant sample (GABA dose: 4.6 × 10^−3^ μmol/kg BW) assessed in the present study lacked antihypertensive effects, as the GABA content in this preparation was below the assumed effective dose. In addition to ACh and GABA, chlorogenic acid has been identified as a further functional compound in eggplants that has blood pressure regulatory effects [13]. It has previously been demonstrated that the chlorogenic acid content of eggplants does not change markedly after 15 days of refrigerated storage at 4 °C [30], and thus we assume that the chlorogenic acid content in the ACh-reduced eggplant sample remains relative stable during prolonged storage. This also indirectly confirms that the chlorogenic acid content of eggplants would not reach an effective dose.

In clinical trials, the effective doses of GABA for lowering blood pressure have been found to range from 10 to 80 mg/day [7,16,17], whereas tranquilizer-like effects have been observed at concentrations between 26.4 and 70 mg/day [31,32,33]. Furthermore, the recommended daily intake of GABA in several dietary supplements worldwide is 100 mg, consumed in divided doses [34]. In contrast, it has been established that ACh can contribute to reducing blood pressure and improve psychological status at concentrations as low as 2.3 mg/day [15]. Moreover, given that orally administered ACh is not absorbed but instead lowers blood pressure via the regulatory effects of sympathetic activity, there are no potential side effects from excessive intake [18], thereby indicating ACh to be a superior novel functional compound. In addition, our findings in this study indicate that only the ACh content and not the GABA content needs to be evaluated to determine the hypotensive effects of eggplant, thereby simplifying investigations of the hypotensive effects of different eggplant varieties. This conclusion can be generalized in that it is based on significant results from placebo-controlled parallel-group comparison studies using SHRs, which have several human applications and are highly extrapolatable. However, further research is needed to confirm the validity of this conclusion.

## 4. Materials and Methods

### 4.1. Chemicals

An Arium 611 ultrapure water system (Sartorius, Göttingen, Germany) was used to provide ultrapure water with a specific resistance of 18.2 MΩ × cm.

Reagents used in the ACh and GABA combination study: ACh chloride was obtained from Kanto Chemical Co., Inc. (Tokyo, Japan) and GABA was obtained from Tokyo Chemical Industry Co., Ltd. (Tokyo, Japan). The median lethal dose of ACh and GABA is 2500 mg/kg and 12,680 mg/kg, respectively.

Reagents used for ACh and GABA quantification: methanol (high-performance liquid chromatography grade), 1 mol/L hydrochloric acid (HCl), and formic acid were obtained from Nacalai Tesque, Inc. (Kyoto, Japan).

### 4.2. Animals and Ethics Statement

SHR/Izm rats (male 10 weeks old; Japan SLC, Inc., Shizuoka, Japan) were used in the combined treatment effect experiment (ACh and GABA combination effect study). SHR/NCrlCrlj rats (male 14 weeks old; Charles River Laboratories Japan, Inc., Kanagawa, Japan) were used in tests of a single oral administration and SHR/NCrlCrlj rats (male 10 weeks old; Charles River Laboratories Japan, Inc., Kanagawa, Japan) were used in tests of chronic oral administration (validation experiment of the blood pressure-lowering effect of GABA in eggplant powder containing a hypotensive effective amount of ACh). SHRs were maintained in cages under a 12-h light/dark cycle at 23 ± 4 °C with a humidity of 50% ± 20%. These rats had unrestricted access to faucet water and laboratory feed (MF; Oriental Yeast Co., Ltd., Tokyo, Japan). In total, 54 rats were used in experiments. In the ACh and GABA combination effect study, we used 18 rats, which were allocated to one of the following three groups (*n* = 6 per group): ACh (numbered A1 to A6), GABA (numbered G1 to G6), and ACh + GABA (numbered AG1 to AG6) groups. In the validation experiment of the blood pressure-lowering effect of GABA in eggplant powder containing a hypotensive effective amount of ACh, we used 36 rats. For the measurement of the acute blood pressure-lowering effects, we allocated 18 rats to the following three groups (*n* = 6 per group): negative control, positive control, and subject groups. For the measurement of the chronic blood pressure-lowering effect, we allocated 18 rats to the following three groups (*n* = 6 per group): negative control, positive control, and subject groups. All experimental procedures conducted in this study were approved by The Animal Care Committee of the Faculty of Shinshu University (approval number: 290061).

In addition, we used two strains of SHR in this study. In the ACh and GABA combination efficacy study, we selected SHR/Izm based on the fact that we needed to determine the minimum effective dosages of ACh and GABA, and a literature search revealed that the minimum effective dosages of ACh and GABA have only been assessed in this strain.

In the validation experiment of the blood pressure-lowering effect of GABA in eggplant powder containing a hypotensive effective amount of ACh, we set the ACh and GABA concentrations in the samples based on the ACh content of eggplant in a previous study using SHR/NCrlCrlj rats [13], and thus we selected this strain for consistency.

Regarding the age of the selected experimental animals, that of SHR/Izm was determined based on the variation in blood pressure with age, described in the literature for this strain, which indicated that blood pressure undergoes a significant upward trend from 10 weeks of age [35].

In the validation experiment of the blood pressure-lowering effect of GABA in eggplant powder containing a hypotensive effective amount of ACh, we used rats of 14 weeks of age in the acute antihypertensive effect experiment, which is consistent with that of the animals used in a previous study [13]. We selected rats of 10 weeks of age in the chronic antihypertensive effect trial in order to match the age of the SHR/NCrlCrlj rats at the end of the trial with that of the SHR/NCrlCrlj rats in the acute antihypertensive effect trial.

### 4.3. Sample Preparation

ACh and GABA combination effect study: The dosages administered to rats in the three study groups, namely, ACh, GABA, and ACh + GABA, were 10 μmol/kg BW (ACh group), 5 μmol/kg BW (GABA group), and 10 + 5 μmol/kg BW (ACh + GABA group). The concentrations of GABA administered were determined based on reference to the hypotensive effect dose-dependent range for SHR/Izm rats (0.5–50 μmol/kg BW) and by the findings of our preliminary experiments [36]. ACh concentrations were set at the lowest hypotensive effect dose (10 μmol/kg BW) for SHR/Izm rats [37].

Validation experiment of the blood pressure-lowering effect of GABA in eggplant powder containing a hypotensive effective amount of ACh: Eggplant (Tosataka) fruits were harvested from Kochi Agricultural Research Center (Kochi Prefecture, Japan) in July 2019 for the preparation of an ACh-reduced eggplant sample and in December 2020 for the preparation of an eggplant control sample. The eggplant fruits (edible parts), with the calyx removed, were cut into 2 to 3 cm cubes. In our preliminary experiments, we established that the ACh content in eggplant declined by up to 71% when stored at −20 °C for 12 weeks (Figure 3; Experimental method: 21 samples of Tosataka eggplants were cut into 2 to 3 cm pieces and stored in a freezer at −20 °C. Three samples were taken out at 0, 2, 4, 6, 8, 10, and 12 weeks to quantify the ACh content). Thus, to ensure that the ACh content in eggplant was reduced to a very low level, we extended the storage period by freezing 2 to 3 cm cubes of eggplant at −20 °C for 15 months. The ACh-reduced eggplant material was freeze-dried in a freeze-dryer (FDU-2110; EYELA Tokyo Rikakikai Co., Ltd., Tokyo, Japan) and subsequently powdered using a mill mixer (1 min, 28,000 rpm: Wonder Crusher WC-3; Osaka Chemical Co., Ltd., Osaka, Japan) to produce an ACh-reduced eggplant powder. Eggplant control samples were prepared by cutting eggplant fruits into 2 to 3 cm squares immediately after harvesting and then freeze-drying and powdering as previously described. The lyophilized powders were suspended in pure water to prepare the respective samples. The dosages are shown in Table 3.

### 4.4. Quantification of ACh and GABA in Eggplants

Lyophilized eggplant powder (approximately 10 mg) was added to a 1.5 mL tube containing 50 mmol/L HCl (200 µL), vortex shaken for 3 min, and centrifuged (3 min, 1000× *g*, 25 ± 5 °C) to collect the supernatant. To the remaining residue, we added 200 µL of 50 mmol/L HCl, followed by vortex shaking (3 min), centrifugation (3 min, 1000× *g*, 25 ± 5 °C), and collection of the supernatant. These steps were repeated twice. All collected supernatants were homogeneously mixed (approximately 600 μL) and made up to a final volume of 1 mL with 50% (*v*/*v*) methanol containing 0.01% formic acid, followed by thorough mixing. After filling up to 1 mL, the samples to be measured were diluted 50-fold with the 50% (*v*/*v*) methanol containing 0.01% formic acid and quantification was performed using the standard addition method and liquid chromatography-tandem mass spectrometry (LC-MS/MS).

The quantification of ACh and GABA was performed based on previously described methodology [13]. The analytical systems used were Nexera-I LC-2040C 3D (UPLC) and LCMS-8045 (MS) (Shimadzu Co., Kyoto, Japan) and the column used was a YMC-Triart PFP column (4.6 mm × 250 mm, 5 µm). The chromatographic conditions were as follows: mobile phase of 50% (*v*/*v*) methanol containing 0.01% formic acid at a flow rate of 0.50 mL/min, a separation temperature of 40 °C, an injection volume of 1 µL, and an analysis time of 18 min. Multiple reaction monitoring data were optimized in positive electrospray ionization (ESI) mode (ESI [+] MRM). LC-MS/MS analysis employed multiple reaction monitoring, a desolvation line temperature of 250 °C, an interface temperature of 300 °C, a heat block temperature of 400 °C, drying and heating gases flowed at 10 L/h, and nebulizer gas flowed at 3.0 L/min. The voltage settings were as follows: Q1 Pre-Bias (V) was set to −10.0 (ACh) and −11.0 (GABA); collision energy (V) was set to −14.0 (ACh) and −13.0 (GABA); and Q3 Pre-Bias (V) was set to −17.0 (ACh) and −17.0 (GABA). Monitoring was performed at the following mass-to-charge ratio (*m*/*z*) transitions: 146.15 → 87.10 (ACh) and 104.15 → 87.20 (GABA). The methodology adopted has been described in our previous study, in which we used the standard addition method to measure the concentrations of ACh and GABA [38].

### 4.5. Probability Sum Test

To determine the combined effects of ACh and GABA, we used a modified probability sum test [39] (abbreviated as the “*q* test”), which is based on classic probability analysis and has been suggested to be appropriate for evaluating the synergism of a combination of two medications [39,40]. SHR/Izm rats were acclimated for 1 week prior to taking blood pressure measurements and subsequent grouping, thereby ensuring that there were no significant differences among the groups with respect to baseline blood pressures. The sample size of each group was set to six based on our previous experimental experience [13]. The 10-week-old male SHR/Izm rats were allocated to one of the following three study groups: ACh group (10 μmol/kg BW), GABA group (5 μmol/kg BW), and ACh + GABA group (10 + 5 μmol/kg BW). The rats were initially fasted overnight and at 09:00 the following day were orally administered the indicated samples using a feeding needle (gavage). Prior to and 6 h after oral administration, we measured SBP using the tail-cuff method (BP-98A; Softron Co., Tokyo, Japan). On the basis of clinical experience, the rats were classified as responders when their SBP dropped by more than 20 mmHg [41].

The formula used to determine the *q*-value is as follows:$$q=P_{A+B}/\left(P_{A}+P_{B}-P_{A}\times P_{B}\right)$$
where *P* (probability) represents the percentage of responders in each group (*P* = number of responders/all rats in the group), and *A* and *B* represent samples *A* and *B*, respectively. *P_A_*
_+ *B*_ is the real percentage of the responders in combinations with *A* and *B*. The expected response rate is (*P_A_* + *P_B_* − *P_A_* × *P_B_*), where (*P_A_* + *P_B_*) represents the sum of probabilities when samples *A* and *B* are used alone and (*P_A_* × *P_B_*) is the probability of rats responding to both samples when used alone (i.e., assuming the two medications act independently). In this study, samples *A* and *B* corresponded to ACh and GABA, respectively. The combination was considered antagonistic when *q* < 0.85, synergistic when *q* > 1.15, and additive when *q* = 0.85–1.15 [39,40].

### 4.6. Measurement of the Acute Antihypertensive Effect

SHR/NCrlCrlj rats (male 14 weeks old) were acclimated for 1 week prior to taking blood pressure measurements and subsequent grouping, as described in the previous section. The three assessed groups were as follows: a negative control group (pure water, *n* = 6), a positive control group (eggplant control sample, *n* = 6), and a subject group (ACh-reduced eggplant sample, *n* = 6). Doses were set at 0.073 mg/kg BW (ACh: 2.7 × 10^−12^ mol/kg BW; GABA: 4.6 × 10^−9^ mol/kg BW) of the ACh-reduced eggplant sample and 0.059 mg/kg BW (ACh: 10^−9^ mol/kg BW; GABA: 4.6 × 10^−9^ mol/kg BW) of the eggplant control sample. The eggplant control sample contained an ACh dose of 10^−9^ mol/kg, which is consistent with the dosage used in our previous study [13]. The ACh-reduced eggplant sample was designed to contain the same content of GABA as the eggplant control sample. The rats in each group were fasted overnight and at 09:00 the following day were orally administered the indicated samples with a feeding needle. To determine the changes in blood pressure, SBP was measured prior to and 3, 6, 9, and 24 h after oral administration using the tail-cuff method.

### 4.7. Measurement of the Chronic Antihypertensive Effect

SHR/NCrlCrlj rats (male 10 weeks old) were acclimated for 1 week prior to taking blood pressure measurements and subsequent grouping, as described in Section 4.5. The three assessed groups were as follows: a negative control group (pure water, *n* = 6), a positive control group (eggplant control sample, *n* = 6), and a subject group (ACh-reduced eggplant sample, *n* = 6). Doses were set at 0.73 mg/kg BW (ACh: 2.7 × 10^−11^ mol/kg BW; GABA: 4.6 × 10^−8^ mol/kg BW) of the ACh-reduced eggplant sample and 0.59 mg/kg BW (ACh: 10^−8^ mol/kg BW; GABA: 4.6 × 10^−8^ mol/kg BW) of the eggplant control sample. Given that assessments of the chronic antihypertensive effect were conducted without fasting, the doses of ACh and GABA in chronic oral administration were set at 10-fold higher than those used in single oral administration in order to prevent feed intake adversely influencing absorption of the sample. The oral administration of samples to rats was repeated daily for 28 days. SBP was measured prior to administration and on days 7, 14, 21, and 28 during the administration period using the tail-cuff method. Blood pressure was measured between 09:00 and 12:00, and the indicated samples were orally administered using a feeding needle (gavage) at 18:00 each day. The changes in SBP were defined as the blood pressures measured after oral administration (days 7, 14, 21, and 28) minus the blood pressure measured prior to administration (day 0). Throughout the study period, water and food consumption were assessed twice weekly, and body weights was measured once each week.

### 4.8. Statistical Analysis

All experimental results are presented as the means ± standard error. Differences were considered significant at *p* < 0.05 and highly significant at *p* < 0.01. The *F*-test was used to confirm the homogeneity of variance of the data. Student’s *t*-test was used to compare the means of groups with homoscedastic data, whereas Welch’s *t*-test was used to compare the means of groups with heteroscedastic data. Analyses were performed using Microsoft Excel 365 MSO 2212 Build 16.0.15928.20196.

## 5. Conclusions

In this study, we evaluated the effects of GABA on the hypotensive effects of eggplant using SHRs and confirmed the main constituents associated with the antihypertensive effects of eggplants. The results of the combined effect study of ACh and GABA revealed that the effects of ACh and GABA were additive rather than synergistic. To assess the antihypertensive effects of GABA in eggplant, we orally administered SHRs with ACh-reduced eggplant samples containing almost no ACh but unaltered levels of GABA, along with eggplant control samples containing ACh and GABA, to examine blood pressure changes. The results revealed that only the eggplant control samples had an antihypertensive effect. On the basis of these findings, we speculate that in clinical studies, GABA makes a negligible contribution to the detected antihypertensive effects of eggplant, as the contents of GABA in eggplant powder are less than the effective dose and this compound shows no synergistic effect with ACh. Accordingly, we concluded that only the ACh content needs to be considered when establishing an appropriate beneficial intake of eggplant.

Overall, the blood pressure-lowering dose of ACh is lower than that of GABA, a functional compound commonly used for its blood pressure-lowering effects. Moreover, ACh is effective in small amounts without being absorbed by the body. Consequently, ACh is considered a promising novel and safe functional food constituent. On the basis of our findings in this study, we intend to further assess efficacious use of foods containing the functional constituent ACh and to search for other promising ACh-containing agricultural, forestry, and fishery food resources, thereby contributing to a healthier society through the diversification of ACh intake options. In this study, we focused on evaluating the effects of the GABA content of eggplant on its antihypertensive effects; however, eggplants also contain other antihypertensive compounds, such as chlorogenic acid, and although the levels of chlorogenic acid in eggplants do not appear to reach those considered necessary for an appreciable therapeutic effect, its plausible influence on the antihypertensive effects of eggplant needs to be further investigated (for example, to establish whether it has any synergistic effects with ACh).

## Figures and Tables

**Figure 1 molecules-28-02835-f001:**
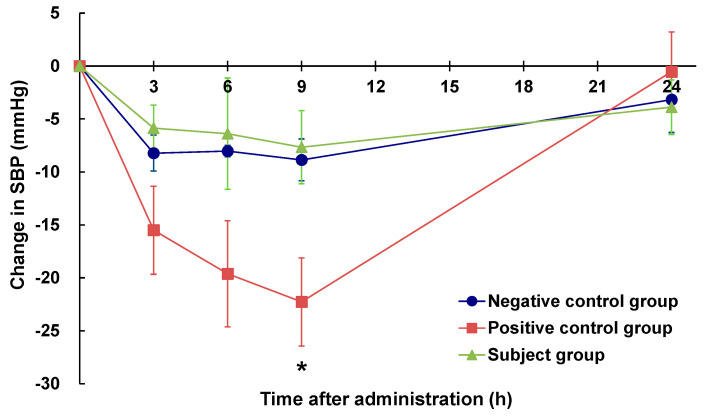
Changes in the SBP of SHRs after a single oral administration of eggplant suspension. SBP: systolic blood pressure. ●: negative control group (*n* = 6), ■: positive control group (*n* = 6), ▲: subject group (*n* = 6). The negative control group was administered pure water, the positive control group was administered an eggplant control sample, and the subject group was administered an ACh-reduced eggplant sample. * *p* < 0.05 versus the negative control group, evaluated using Student’s *t*-test (homoscedasticity) and Welch’s *t*-test (heteroscedasticity).

**Figure 2 molecules-28-02835-f002:**
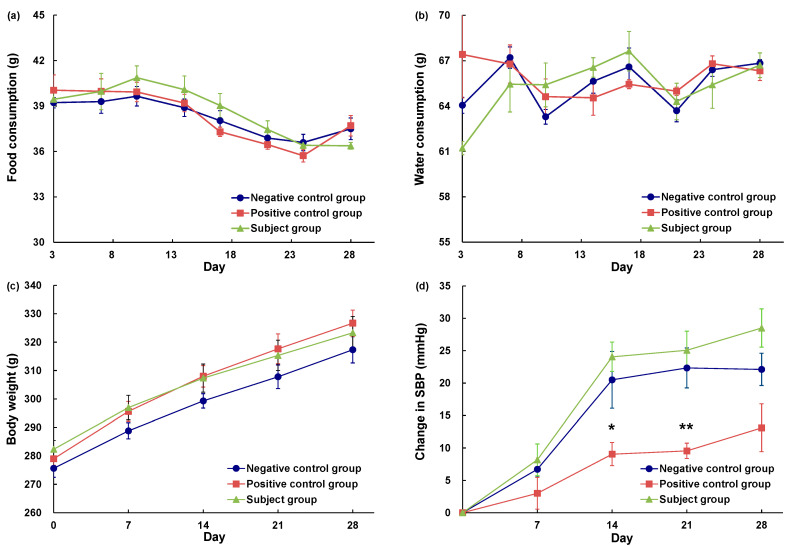
Daily food consumption (**a**), water consumption (**b**), body weight (**c**), and change in the SBP (**d**) of SHRs during chronic administration of eggplant powder suspension. SBP: systolic blood pressure. ●: negative control group (*n* = 6), ■: positive control group (*n* = 6), ▲: subject group (*n* = 6). The negative control group was administered pure water, the positive control group was administered an eggplant control sample, and the subject group was administered an ACh-reduced eggplant sample. * *p* < 0.05, ** *p* < 0.01 versus the negative control group, evaluated using Student’s *t*-test (homoscedasticity) and Welch’s *t*-test (heteroscedasticity).

**Figure 3 molecules-28-02835-f003:**
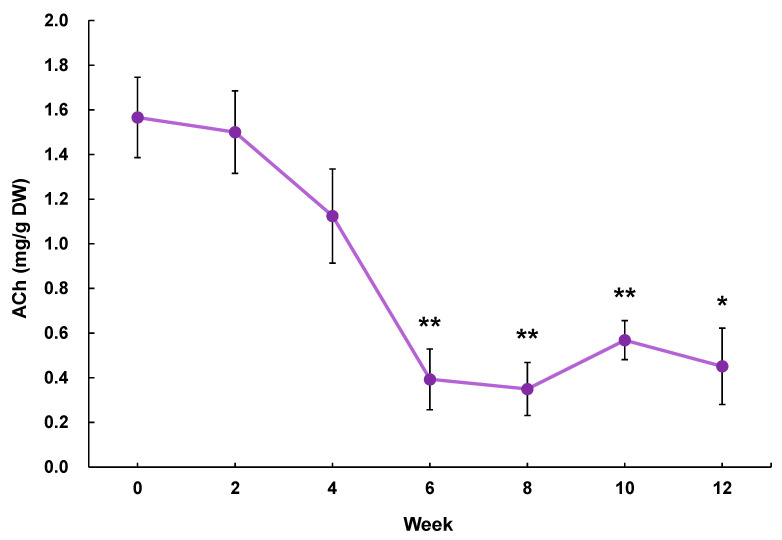
Changes in the ACh content in eggplant (Tosataka) fruit stored at −20 °C. * *p* < 0.05, ** *p* < 0.01 versus week 0, evaluated using Student’s *t*-test. ACh: acetylcholine.

**Table 1 molecules-28-02835-t001:** The *q* test results for SHRs following acute ACh and GABA treatment (*n* = 6 in each group).

Group	Medication	Dose (μmol/kg BW)	Identification Number	SBP (mmHg)			Probability	*q*-Value
				Before	After	Change		
ACh	ACh	10	A1	178	157	−21	1/3	0.90
			A2	184	176	−9		
			A3	186	163	−23		
			A4	187	206	19		
			A5	195	191	−5		
			A6	168	161	−7		
GABA	GABA	5	G1	179	170	−8	1/3	
			G2	170	186	16		
			G3	190	169	−22		
			G4	157	147	−11		
			G5	188	166	−22		
			G6	157	145	−11		
ACh + GABA	ACh + GABA	10 + 5	AG1	188	194	6	1/2	
			AG2	191	176	−15		
			AG3	176	175	−1		
			AG4	176	144	−32		
			AG5	191	171	−21		
			AG6	206	165	−41		

ACh: acetylcholine; GABA: γ-aminobutyric acid; BW: body weight; SBP: systolic blood pressure. *q* = *P_ACh_*
_+ *GABA*_/(*P_ACh_* + *P_GABA_* − *P_ACh_* × *P_GABA_*); *P_ACh_*: probability of the ACh group, *P_GABA_*: probability of the GABA group, *P_ACh_*
_+ *GABA*_: probability of the ACh + GABA group.

**Table 2 molecules-28-02835-t002:** Contents of ACh and GABA in eggplant samples (mg/g DW; *n* = 3).

Sample	ACh	GABA
ACh-reduced eggplant sample (ACh-reduced eggplant powder)	0.0055 ± 0.00070	6.5 ± 0.28
Eggplant control sample (unmodified eggplant powder)	2.5 ± 0.12	8.1 ± 0.10

ACh: acetylcholine; GABA: γ-aminobutyric acid; DW: dry weight.

**Table 3 molecules-28-02835-t003:** Dose setting for validation experiment of the blood pressure-lowering effect of GABA in eggplant powder containing a hypotensive effective amount of ACh.

Experiment	Sample	Dose (µg/kg BW)	ACh (mol/kg BW)	GABA (mol/kg BW)
Single oral administration	Eggplant control sample	59	1.0 × 10^−9^	4.6 × 10^−9^
	ACh-reduced eggplant sample	73	2.7 × 10^−12^	
Chronic oral administration	Eggplant control sample	590	1.0 × 10^−8^	4.6 × 10^−8^
	ACh-reduced eggplant sample	730	2.7 × 10^−11^	

ACh: acetylcholine; GABA: γ-aminobutyric acid; BW: body weight.

## Data Availability

The submitted manuscript contains all of the data, models, and codes generated or utilized during this study.

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
