# Peer review of "Evaluation of the Antihypertensive Activity of Eggplant Acetylcholine and γ-Aminobutyric Acid in Spontaneously Hypertensive Rats"

_molecules, 2023, doi:10.3390/molecules28062835_

Round 1
Reviewer 1 Report
Related works have already been published, I think by the same authors also. Even though this manuscript lacks novelty, it is well drafted. The abstract section needs revision, results, methodology and discussion parts are fine.
1. The abstract does not adequately describe the essence of study nor does it arouse interest in readers to read the manuscript. it will be good if you can modify the abstract.
2. In the Introduction part, the main purpose of the study as well as study objectives should be more highlighted. The authors must justify this research work since they published similar works before.
3. Provide collection details of Solanum melongena? Do the authors cultivated the plant sample or whether they procured from other sources?
4. Lines 98-99; To reduce only ACh, 2–3 cm cubes of egg- 98 plant were frozen at -20°C for 15 months - provide reference
5. Line 283; It is stated that the major antihypertensive compound in eggplant is ACh for the first time. But it is already reported, I think by same authors in Food Chemistry, Vol: 276. (https://www.sciencedirect.com/science/article/pii/S0308814618317825)
6. Include few lines on the limitations of your study in the conclusion section.
Author Response
- The abstract does not adequately describe the essence of study, nor does it arouse interest in readers to read the manuscript. it will be good if you can modify the abstract.
Response: Thank you for your suggestions. We have rewritten the abstract in line with the comments (Line 16-30).
- In the Introduction part, the main purpose of the study as well as study objectives should be more highlighted. The authors must justify this research work since they published similar works before.
Response: In the revised manuscript, we have modified the Introduction as suggested (Line 78-88).
- Provide collection details of Solanum melongena? Do the authors cultivated the plant sample or whether they procured from other sources?
Response: Eggplant (Tosataka) fruits were harvested from Kochi Agricultural Research Center (Kochi Prefecture, Japan) in July 2019 for preparation of the ACh-reduced eggplant sample and in December 2020 for preparation of the eggplant control sample. We have provided the indicated information in the revised manuscript (Line 320-323).
- Lines 98-99; To reduce only ACh, 2–3 cm cubes of eggplant were frozen at -20°C for 15 months – provide reference
Response: In our preliminary experiments, we established that the ACh content in eggplants declines by 71% when stored at -20°C for 12 weeks. Consequently, we extended the storage period to 15 months in order to ensure that the ACh content in eggplant is extremely low. We have included this information in the Materials and Methods of the revised manuscript (Lines 323–330).
- Line 283; It is stated that the major antihypertensive compound in eggplant is ACh for the first time. But it is already reported, I think by same authors in Food Chemistry, Vol: 276. (https://www.sciencedirect.com/science/article/pii/S0308814618317825)
Response: We have revised the relevant description to indicate this (Lines 160–166).
- Include few lines on the limitations of your study in the conclusion section.
Response: In accordance with the suggestion, we have added relevant information regarding the limitations of the present study in the Conclusion (Lines 463–469).

Reviewer 2 Report
In this manuscript, the authors indicate evaluation of Antihypertensive Activity of Acetylcholine and 2 γ-Aminobutyric Acid in Eggplant. The manuscript can be accepted after addressing the below mentioned corrections.
1. A little more information about antihypertensive and acetylcholine can be given in the introduction. For this purpose the authors can look at the following articles for introduction section:Journal of biochemical and molecular toxicology 33 (6), e22313, Journal of biochemical and molecular toxicology 33 (10), e22381.
2. How were the concentrations given to the mice determined? Has an LD study been done?
3. When the prepared plant extract was dissolved in water, was the substance completely dissolved? Didn't it crash? In general, plant extracts dissolve better in organic solvents such as ethanol.
4. There are several English language issues. It should be corrected.
Author Response
- A little more information about antihypertensive and acetylcholine can be given in the introduction. For this purpose, the authors can look at the following articles for introduction section: Journal of biochemical and molecular toxicology 33 (6), e22313, Journal of biochemical and molecular toxicology 33 (10), e22381.
Response: Thank you for your suggestions and for drawing our attention to the relevant literature; accordingly, we have modified the text in the Introduction (Line 35-44, 51-61).
- How were the concentrations given to the mice determined? Has an LD study been done?
Response: We apologize for the lack of clarity in the details of the study.
- Regarding the dosage of ACh and GABA
- ACh and GABA combination effect
GABA concentrations were determined by reference to the hypotensive effect dose-dependent range for SHR/Izm (0.5–50 μmol/kg BW) and on the basis of the findings of preliminary experiments [1]. ACh concentrations were set at the lowest hypotensive effect dose (10 μmol/kg BW) for SHR/Izm rats [2]. (Lines 314–318)
- Validation experiment of the blood pressure lowering effect of GABA in eggplant powder containing a hypotensive effective amount of ACh.
- Acute antihypertensive experiment
For the eggplant control sample, we first quantified the contents of ACh and GABA in eggplant powder by LC-MS/MS, and then determined the amount of eggplant powder used according to the dose of ACh (1.0 × 10-9 mol/kg BW). The dose of GABA (4.6 × 10-9 mol/kg BW) was calculated based on the GABA content in the eggplant powder used. The dose of ACh (1.0 × 10-9 mol/kg BW) was decided with reference to our previous study [3].
For the ACh-reduced eggplant sample, we first quantified the content of ACh and GABA in eggplant powder by LC-MS/MS, and then determined the amount of eggplant powder used according to the GABA dose of 4.6 × 10-9 mol/kg BW, which the same of the eggplant control sample. The dose of ACh (2.7 × 10-12 mol/kg BW) was calculated based on the ACh content in the eggplant power used (Lines 408–409).
- Chronic antihypertensive effect experiment
The concentration used in the chronic antihypertensive effect experiment was 10-fold higher than that used in the acute antihypertensive effect experiment in order to eliminate the effects of feed intake on the absorption of samples, as the chronic antihypertensive effect experiment was conducted without an initial fasting (Lines 422–426).
- About LD study
We did not conduct an LD study. However, studies on the LD50 of ACh and GABA have been provided as 2500 mg/kg and 12680 mg/kg by each safety data sheet and were added to section 4.1. Chemicals (Line 266).
According to Cayman Chemical's Safety Data Sheet (page 7), the LD50 value of ACh is 2500 mg/kg (oral) (https://cdn.caymanchem.com/cdn/msds/23829m.pdf).
According to the safety data sheet of Oxford Lab Chem LLP (page 1), the LD50 value of GABA is 12680 mg/kg (oral) (https://www.oxfordlabfinechem.com/msds/(A-00652)%20GAMA%20AMINO%20BUTYRIC%20ACID%20(GABA)%20(P-Amino%20Butyric%20Acid).pdf).
References
- Hayakawa, K.; Kimura, M.; Kasaha, K.; Matsumoto, K.; Sansawa, H.; Yamori, Y. Effect of a γ-aminobutyric acid-enriched dairy product on the blood pressure of spontaneously hypertensive and normotensive Wistar–Kyoto rats. J. Nutr. 2004, 92, 411–417.
- Yamaguchi, S.; Matsumoto, K.; Wang, W.; Nakamura, K. Differential antihypertensive effects of oral doses of acetylcholine between spontaneously hypertensive rats and normotensive rats. Foods. 2021, 10, 2107.
- Yamaguchi, S.; Matsumoto, K.; Koyama, M.; Tian, S.; Watanabe, M.; Takahashi, A.; Miyatake, K.; Nakamura, K.; Nakamura, K. Antihypertensive effects of orally administered eggplant (Solanum melongena) rich in acetylcholine on spontaneously hypertensive rats. Food Chem. 2019, 276, 376–382.
- When the prepared plant extract was dissolved in water, was the substance completely dissolved? Didn't it crash? In general, plant extracts dissolve better in organic solvents such as ethanol.
Response: To address this problem, we took measures to ensure that the material was thoroughly suspended prior to administration (including shaking and ultrasound treatments). Moreover, given that ACh and GABA are generally characterized by good water solubility, we believe that the vast majority of the ACh and GABA in sample preparations was dissolved. In addition, as these samples were administered orally, we wished to avoid the use of organic solvents to prevent any unnecessary stimulation that could contribute to abnormal changes in blood pressure.
- There are several English language issues. It should be corrected.
Response: As requested, the revised manuscript had been checked for English language-related issues.

Reviewer 3 Report
Overview of the manuscript
The manuscript is focuses on the analysis of the antihypertensive effect of Acethylcholine (ACh) and GABA contained in eggplant. In particular, the authors investigated the differential or combined effect of ACh and GABA, using the hypertensive animal model of SHR.
The authors observe that the ACh contained in eggplant is the effective compound able to reduce hypertension, while the GABA contained in eggplant is able to induce a strengthening of the anti-hypertensive effect of ACh, probably acting on different biological mechanism compared to ACh.
GENERAL COMMENT
The work is interesting, the experimental plan is well organized and performed to achieve the aim of the work. The methods used are consistent with the main topic of the work and support the results and conclusions. The statistical procedure is adequate. However, the work should be extensively revised in the presentation form. Several points remain not well described or not clear.
SPECIFIC COMMENTS
Title: Title should be changed. It does not represent the real topic of the work, you investigated ACh and GABA contained in eggplants, not in eggplants, but in hypertensive animal model.
Abstract
Pag. 1, line 18-19: The sentence remains not clear. Rewrite it.
Materials and Methods
Pag. 2, line 69 – 73: You have taken under study two sub-races of SHR. You should explain the reason of this choice and the different ages of the animals used.
Pag. 3, line 98 – 99: Do you have any reference for this procedure? Or is it your own experimental findings obtained previously. Give indication.
Results
Pag. 6, line 225 – 227: the sentence remains not clear. You seems to indicate a contradiction. Rewrite the sentence.
Table 2: your data show a large increase in the blood pressure level in one animal in Ach group, and one in GABA group. Do you have any ideas about this. Give explanations.
Discussion
The conclusions section is just a broad summary of the same concepts developed in Discussion section. In this form is useless for the reader. Change to a more schematic form or delete the section.
Author Response
SPECIFIC COMMENTS
Title: Title should be changed. It does not represent the real topic of the work, you investigated ACh and GABA contained in eggplants, not in eggplants, but in hypertensive animal model.
Response: Thank you for highlighting this point; the title of the revised manuscript has been modified to “Evaluation of the Antihypertensive Activity of Eggplant Acetylcholine and γ-Aminobutyric Acid in Spontaneously Hypertensive Rats.”
Abstract
Pag. 1, line 18-19: The sentence remains not clear. Rewrite it.
Response: We have rewritten the indicated section of the abstract for better clarity (Line 16-30).
Materials and Methods
Pag. 2, line 69 – 73: You have taken under study two sub-races of SHR. You should explain the reason of this choice and the different ages of the animals used.
Response:
- In this study we selected two strains of SHR according to the needs of the experiment. The reasons are as follows.
- ACh and GABA combination effect study.
We selected the SHR/Izm strain, as we wished to determine the minimum effective dosages of ACh and GABA. On the basis of a literature search, we established that such information is available only for SHR/Izm rats. Therefore, we selected this strain in the ACh and GABA combination effect study.
- Validation experiment of the blood pressure lowering effect of GABA in eggplant powder containing a hypotensive effective amount of ACh.
In previous studies on the antihypertensive effect of ACh, the SHR/NCrlCrlj strain rats have been used [1,2], thus in the present study we selected the strain for consistency.
These are the reasons why we use SHR/Izm and SHR/NCrlCrlj.
- Regarding the age of the selected experimental animals
- ACh and GABA combination effect study.
SHR/Izm rats was determined based on the variation in blood pressure with age described in the literature, according to which blood pressure is characterized by a significant upward trend from 10 weeks of age [3].
- Validation experiment of the blood pressure lowering effect of GABA in eggplant powder containing a hypotensive effective amount of ACh.
According to the previous survey, the blood pressure of SHR/NCrlCrlj was significantly higher than that of normotensive WKY/NCrlCrlj rats at the age of 10 weeks, and the rise of blood pressure gradually stabilize after 10 weeks of age [4]. Therefore, we believe that whether 10-week-old or 14-week-old would not affect the consistency of the experiment.
The reasons for choosing the age of 14 weeks for the acute antihypertensive experiment and 10 weeks for the chronic antihypertensive experiment are as follows. In the acute hypotensive effect experiments, we used the same 14-week-old animals as in previous studies [1], whereas in the chronic antihypertensive effect trial, we selected 10-week-old rats, in order to match the age of the SHR/NCrlCrlj rats at the end of the trial with that of the SHR/NCrlCrlj rats used in the acute antihypertensive effect trial. This information has accordingly been included in section 4.2 of the revised manuscript (Lines 293–310).
References
- Yamaguchi, S.; Matsumoto, K.; Koyama, M.; Tian, S.; Watanabe, M.; Takahashi, A.; Miyatake, K.; Nakamura, K.; Nakamura, K. Antihypertensive effects of orally administered eggplant (Solanum melongena) rich in acetylcholine on spontaneously hypertensive rats. Food Chem. 2019, 276, 376–382.
- Yamaguchi, S.; Hayasaka, Y.; Suzuki, M.; Wang, W.; Koyama, M.; Nagasaka, Y.; Nakamura, K. Antihypertensive mechanism of orally administered acetylcholine in spontaneously hypertensive rats. Nutrients. 2022, 14, 905.
- Fukuda, S.; Tsuchikura, S.; Iida, H. Age-related changes in blood pressure, hematological values, concentrations of serum biochemical constituents and weights of organs in the SHR/Izm, SHRSP/Izm and WKY/Izm. Anim. 2004, 53, 67–72.
- Zhang, R., Inagawa, H., Kazumura, K., Tsuchiya, H., Miwa, T., Morishita, N., ... & Soma, G. I. Evaluation of a hypertensive rat model using peripheral blood neutrophil activity, phagocytic activity and oxidized LDL evaluation. Anticancer research. 2018, 38(7), 4289-4294.
Pag. 3, line 98 – 99: Do you have any reference for this procedure? Or is it your own experimental findings obtained previously. Give indication.
Response: This procedure is based on our preliminary studies, in which we detected a 71% reduction in the ACh content of eggplants when stored at -20°C for 12 weeks. Consequently, we extended the storage period to 15 months in order to ensure that the ACh content in eggplants was extremely low. We have included this information in the Materials and Methods of the revised manuscript (Lines 320-323).
Results
Pag. 6, line 225 – 227: the sentence remains not clear. You seems to indicate a contradiction. Rewrite the sentence.
Response: We have revised this sentence for better clarity (Lines 101–104).
Table 2: your data show a large increase in the blood pressure level in one animal in Ach group, and one in GABA group. Do you have any ideas about this. Give explanations.
Response: The reason for that is not clear. However, on the basis of clinical experience and the findings of previous studies, we do not consider a change in SBP greater than 20 mmHg to be abnormal, and thus we believe that the increases in blood pressure seen in the A4 and G2 individuals are within the range of normal blood pressure changes in SHR.
Discussion
The conclusions section is just a broad summary of the same concepts developed in Discussion section. In this form is useless for the reader. Change to a more schematic form or delete the section.
Response: We have rewritten the conclusion as indicated (Lines 160-166).

Round 2
Reviewer 2 Report
The manuscript can be accepted this form.
Reviewer 3 Report
No more concerns.